# Effect of Hygiene Protocols on the Mechanical and Physical Properties of Two 3D-Printed Denture Resins Characterized by Extrinsic Pigmentation as Well as the Mixed Biofilm Formed on the Surface

**DOI:** 10.3390/antibiotics12111630

**Published:** 2023-11-17

**Authors:** Adriana Barbosa Ribeiro, Beatriz Marcatto Tinelli, Lorena Mosconi Clemente, Beatriz de Camargo Poker, Viviane de Cássia Oliveira, Evandro Watanabe, Cláudia Helena Silva-Lovato

**Affiliations:** 1Department of Dental Materials and Prosthesis, Ribeirão Preto School of Dentistry, University of São Paulo, Café Avenue S/N, Ribeirão Preto 14040-904, SP, Brazil; driribeiro@usp.br (A.B.R.); beatriz_tinelli@alumni.usp.br (B.M.T.); lorena.clemente@usp.br (L.M.C.); beatrizpoker@usp.br (B.d.C.P.); vivianecassia@usp.br (V.d.C.O.); 2Department of Restorative Dentistry, Ribeirão Preto School of Dentistry, University of São Paulo, Café Avenue S/N, Ribeirão Preto 14040-904, SP, Brazil; ewatabane@usp.br

**Keywords:** denture, acrylic resin, CAD/CAM, disinfection, properties

## Abstract

To assess the effect of hygiene protocols and time on the physical–mechanical properties and colony-forming units (CFU) of *Candida albicans*, *Staphylococcus aureus*, and *Streptococcus mutans* on 3D-printed denture resins (SmartPrint and Yller) with extrinsic pigmentation compared to conventional resin (CR). The protocols were evaluated: brushing (B), brushing and immersion in water (W), 0.25% sodium hypochlorite (SH), and 0.15% triclosan (T), simulating 0, 1, 3, and 5 years. The data were analyzed by ANOVA with repeated measurements, ANOVA (Three-way) and Tukey’s post-test, generalized linear model with Bonferroni adjustment, and ANOVA (Two-way) and Tukey’s post-test (α = 0.05). The protocols influenced color (*p* = 0.036) and Knoop hardness (*p* < 0.001). Surface roughness was influenced by protocols/resin (*p* < 0.001) and time/resin (*p* = 0.001), and flexural strength by time/protocols (*p* = 0.014). *C. albicans* showed interactions with all factors (*p* = 0.033). *Staphylococcus aureus* was affected by protocols (*p* < 0.001). *Streptococcus mutans* exhibited no count for SH and T (*p* < 0.001). Yller resin showed more color changes. The 3D-printed resins displayed lower microhardness, increased roughness, and decreased flexural strength compared to CR with all protocols in a simulated period of 5 years. The indication of printed resins should be restricted to less than 3 years.

## 1. Introduction

Edentulism is a public health problem that impacts an individual’s physical and mental health. To mitigate this problem, an option is oral rehabilitation with complete dentures, whether conventional or retained by implants. Several clinical consultations are necessary to manufacture a complete denture, which can be a problem for both the professional and the patient. The material used is polymethylmethacrylate (PMMA) and, although it meets the objectives of complete edentulous rehabilitation, there can be residual methyl methacrylate monomer, a change in properties influenced by hygiene methods [1,2,3]. However, the result is dependent on the knowledge of the dental surgeon.

Aiming at reducing the number of clinical visits, predicting results, simplifying laboratory work, and improving the quality of denture devices, Computer-Aided Manufacturing (CAD/CAM) can be used for complete denture manufacturing [2,3,4], either through the subtractive technique performed by milling processed PMMA blocks, which is a polyester derived from methyl methacrylate with only one C=C group and can be polymerized by free radical polymerization process, or by 3D printing materials—one photosensitive resin composed of a photosensitive prepolymer, a reactive monomer, a photoinitiator, and other additives [4,5,6,7,8,9]. Three-dimensional printing represents a method of lower-cost processing [10]. It should be noted that the dentures obtained both by the addition and subtraction techniques may have compromised aesthetics, which can be overcome by characterizing the bases with resinous pigments. Some studies present the results of the behavior of printed resins after exposure to pigmented solutions, such as coffee and red wine [11,12]. However, literature is scarce regarding the behavior of base resins characterized and exposed to long periods of hygiene protocols [13,14].

In the last 10 years, there has been an exponential growth in the number of studies comparing the physical and mechanical properties of 3D-printed resins, pre-polymerized blocks, and conventional resins [6,15,16,17,18,19]. Although the results are promising concerning mechanical and surface properties compared to conventional resins, further long-term follow-up studies are required [3,16]. In addition, some studies have evaluated the effects of chemical disinfectants on biofilm control [2,20,21] and the properties of 3D-printed resin. These studies are important because biofilm control is essential for maintaining oral health. However, the literature shows that effective biofilm removal is accomplished by combining mechanical and chemical methods. Mechanical hygiene intervention can be more effective in removing plaque and organic detritus, and chemical hygiene can contribute to surface disinfection through a reduction of pathogenic microorganisms, mainly *Candida* spp. [22]. Therefore, evaluating the long-term effects of the mechanical method associated with the chemical method is necessary to obtain results that can help predict the durability and quality of denture manufacturing with 3D-printed denture base resin. Hygiene protocols can have adverse effects on denture materials, and in vitro and clinical studies must be performed to ascertain their performance in the long term [23,24,25].

The associated method should consist of brushing using a toothbrush with toothpaste or soap. It is an effective, affordable method and capable of mechanically removing the biofilm. The chemical method consists of the immersion of the dentures in chemical substances capable of disorganizing the biofilm and killing the microorganisms [22,25]. Among the evaluated solutions, those based on 0.25% sodium hypochlorite and 0.15% Triclosan have shown satisfactory results and are clinically indicated [25].

Although current research points to the advantages of complete dentures made using the CAD/CAM system with 3D-printed resin compared to those made conventionally, information about the material’s reactions when exposed to hygiene methods is scarce, and thus long-term studies are needed. Therefore, this study evaluated the effect of four hygiene protocols on the color stability, hardness, roughness, surface characteristics, and flexural strength, as well as the microbial load of a multispecies biofilm composed of *C. albicans*, *S. mutans*, and *S. aureus* formed on two 3D-printed resins with extrinsic surface pigmentation compared to a conventional thermally-cured acrylic resin. The tested hypothesis was that the response variables would be different among the resins after using the hygiene protocols at different times.

## 2. Results

Color change was influenced by the interaction among protocol × material × time (*p* = 0.036). The greatest color changes (ΔE) were observed with the Yller (YL) resin after the simulation of 3 and 5 years of exposure to the sodium hypochlorite (SH) protocol when compared to the other resins. The color change was noticeable after 1, 3, and 5 years of using the SH protocol (NBS scale) [26]. The Smart Print (SP) resin underwent greater color change (ΔE) [26,27] when associated with the SH protocol (Table 1).

The Knoop microhardness was influenced by the interaction between protocol × material × time (*p* < 0.001). The printed resins showed a significant difference compared to the conventional resin (CR) in all protocols and times, except for SP, which was similar to CR after 1 year of brushing and immersion in 0.15% Triclosan (T). At the end of 5 years, the printed resins showed similar hardness after exposure to brushing and immersion in water (W), SH, and T (Table 2).

For surface roughness, the effect of the interaction between resin × hygiene protocol (*p* < 0.001) and time × resin (*p* = 0.001) can be verified (Table 3). The Yller resin showed a reduction in roughness after exposure to the T protocol and intermediate roughness after the SH protocol. The roughness of the SP resin was affected by the SH and T protocols. The printed resins showed similar roughness after using the T protocol. The Yller resin showed stability after all times, and the SP resin had less surface roughness after T5 and intermediate roughness at T3. The conventional resin showed the lowest roughness values, regardless of the time.

Flexural strength was influenced by the interaction between time × hygiene protocol (*p* = 0.014; Table 4). After 5 years, there was a significant reduction in flexural strength after using all protocols, with the SH and T protocols promoting a significant reduction compared to the controls, which were similar to each other.

A comparison of the images before exposure of the specimens to the hygiene protocols showed that the extrinsic pigmentation of SP and YL generated bubbles on the surface and irregularities. No changes such as scratches, grooves, and loss of shine were detected due to hygiene protocols (Figure 1).

In regard to the effect of hygiene protocols on the microbial load for *C. albicans*, there was an interaction between the factors (*p* = 0.033; Table 5), with the YL resin showing a higher CFU count compared to the T hygiene protocol. For CR and SP resins, all protocols were different, with T being the most efficient, followed by brushing and immersion in water (W). For *S. aureus*, the CFU count was influenced only by the protocols (*p* < 0.001), with the best effectiveness being the T protocol, followed by the W protocol (Table 6). For these two species, the SH protocol reduced the microorganism count to zero. For *S. mutans*, the SH and T protocols also reduced the microorganism count to zero. The statistics indicated significance only for the group factor (*p* < 0.001), with a greater reduction in the CFU count promoted by the W protocol (4.10 ± 0.83) compared to the group without hygiene (5.95 ± 0.64).

## 3. Discussion

The hypothesis of the study must be accepted since there was a difference in response variables of the tested resins in regard to the effects of hygiene protocols and time. Hygiene protocols and time of use affected the evaluated properties and the microbial load, reinforcing the need to observe the best combinations between protocol, time of use, and type of resin to be indicated. This finding is in accordance with the literature [2].

The results indicated that the protocol using immersion in sodium hypochlorite at 0.25% generated a significant color change of the resins after the simulation of 1, 3, and 5 years. This result corroborates the literature [2,20]. The change in the optical characteristics of the resins can occur due to the dissociation of chloride ions, which are extremely reactive and may cause surface changes and stains, which is a disadvantage of this solution [2]. Despite these changes, it should be noted that clinically perceptible color changes were identified with Clássico and Yller resins after 3 years of brushing and immersion in water and with the Yller resin in all the periods and hygiene protocols evaluated. Therefore, the results suggest that, in terms of color, the SmartPrint resin showed better clinical results with all hygiene protocols, which makes it a promising material for the area of application.

Dentures made of a material with low surface hardness can be damaged by mechanical brushing, causing plaque retention and pigmentations, which can decrease the life of dentures [5]. Prpić et al. [5] evaluated the mechanical properties (flexural strength and surface hardness) of different materials and technologies for denture base fabrication. The study emphasized the digital technologies of computer-aided design/computer-aided manufacturing (CAD/CAM) and 3D printing. The authors highlight that 3D-printed materials for denture bases are a new option; however, they have lower mechanical properties than most other denture base materials. This is because acrylic resins for 3D printing of removable dentures have relatively low double-bond conversion compared with traditional acrylic resins, which can also affect mechanical properties [5,24].

This study found that the surface hardness of all the resins decreased after immersion in all the denture cleansers and that there was a time-dependent decrease, while the 3D-printed resin showed inferior hardness values compared to the heat-polymerized resin.

This study found that the surface hardness of all resins decreased after immersion in all denture cleaning products and that there was a time-dependent decrease. It is worth mentioning that the reduction in hardness of the printed resin was bigger than that of the heat-polymerized resin. This result is in accordance with other studies [2,6]. The SmartPrint resin showed results similar to those of the Clássico resin after 1 year of exposure to the protocol with immersion in Triclosan 0.15%. In the other conditions, the printed resins presented lower hardness than the heat-polymerized resin [10]. The reduced surface hardness of the 3D-printed resin may be a result of the considerably lower degree of conversion [6,24]. It is not only physical–chemical aspects that lead to this decrease, but also biological factors, such as salivary enzymes and oral microbiota [1]. The latter were not reproduced in this study, representing a methodological limitation.

In this study, the heat-polymerized resin showed significantly lower surface roughness values compared to 3D printed resins, in line with the literature [7,10,21]. The SH and T hygiene protocols influenced the surface roughness of the Yller and SmartPrint printed resins, but the Yller resin became stable over time and the SP resin showed a reduction of the surface roughness with the simulation of 3 and 5 years of immersion. The results can be justified based on the initial periods in which the surface may have been affected by the oxidizing effect of chemical solutions [23], which then stabilized over time. The literature shows that combined methods of hygiene with brushing and immersion with disinfectant solutions are more effective for biofilm control [22], but there is limited evidence regarding the effect of denture hygiene interventions on the physical and mechanical properties of the denture base materials [25].

The characterization of the surface of the printed acrylic resins can explain the increased roughness values compared to the heat-polymerized resin, which are in accordance with the findings of the qualitative analysis of the surfaces of the specimens by Confocal Laser Microscopy. However, to our knowledge these results disagree with Tasin et al. [4] in that only one study evaluated the application of characterizing agents on the surface of acrylic resin, and the results indicated that the union of these materials with the printed resins needs to be improved [13].

Microbial colonization [15,28,29] and color stability can be negatively affected by this increased surface roughness, requiring the user to clean more vigorously. This can result in minor variations in surface conditions [15,17,18]. Thus, the results of the present study suggest that printed resins subjected to extrinsic characterization to obtain adequate aesthetics are more susceptible to this colonization.

Although there was no relevant difference between the resins [3], the flexural strength significantly decreased with the hygiene protocols after 5 years. This suggests that, in addition to cleaning solutions, the mechanical method of cleaning and the time of use can reduce the resistance of these materials. This is consistent with the study by Abualsaud and Gad [19], in which the additive CAD/CAM method was comparable to the conventional thermopolymerization technique. However, Prpić et al. [5] reported the highest values with milled PMMA, followed by heat polymerization and, lastly, 3D-printed PMMA. In the studies by Fouda et al. [7] and Freitas et al. [10], 3D-printed resins also had the lowest flexural strength.

The protocols using immersion in sodium hypochlorite at 0.25% and triclosan at 0.15% [28] were similar to each other, and there was a decrease in flexural strength over time when compared to controls. Despite the decrease found, the values respect the recommendations by the ISO standard 20795-1 (65 Mpa) [30]. These results are important since both solutions are efficient for biofilm control and denture-related stomatitis [28].

As previously mentioned, the characterization of the surface of the printed acrylic resins can explain the presence of the generated bubbles and irregularities on the surface compared to the heat-polymerized resin. Moreover, another explanation is the materials’ inner structures and low double-bond conversion of 3D printed resins compared with traditional acrylic resins, affecting mechanical properties [5].

As for the antimicrobial action, the SH protocol was the most efficient, reducing the CFU counts of the three microorganisms to zero, corroborating previous studies [31]. In addition to bactericidal and fungicidal action, 0.25% sodium hypochlorite has low cost and easy implementation [32], and this solution is commonly indicated for biofilm control and cleaning of prosthetic devices [31]. The antimicrobial superiority of this solution is related to its action on essential enzymatic sites of bacteria, promoting irreversible inactivation through hydroxyl and chloramine ions [32,33]. Furthermore, the solution is capable of dissolving organic substances present on acrylic surfaces, such as lipids and fatty acids, reducing the surface tension of the resins [32,33]. The 0.25% concentration was used because it is safe and effective. High concentrations, such as 1%, should not be used because they cause changes in color and a decrease in flexural strength [31].

Triclosan 0.15% was also effective, being able to replace the SH protocol, corroborating other studies [28,31,32]. This solution has shown good antimicrobial action and acceptable effects on complete denture base materials [29]. This is because this substance affects the synthesis of RNA and proteins, being able to cause cell lysis [34,35,36,37]. As there are few studies, it was included as an alternative to sodium hypochlorite, which has an unpleasant odor and can be irritating to tissues [31].

Water was also used, as it is indicated for immersion of prostheses during sleep [29] and prevents the release of ions by the resin [38]. As it does not contain active ingredients in composition, the reduction in CFU count by the W protocol shows that brushing should be indicated as an essential element for cleaning dentures. Although chemical control of biofilm is useful and appropriate, it must always be accompanied by mechanical control [29].

The protocol without brushing, in turn, was used as a control to show biofilm formation, ensuring that the reduction or elimination of microorganisms occurred due to hygiene protocols.

Although there was no difference in the initial biofilm formation, the T protocol was less effective in eliminating *C. albicans* in the YL resin. This may be associated with hydrophobicity and the higher level of mucin on the surface. Mucins adhere to substrates through the hydrophobic protein portion of their macromolecule, leaving their carbohydrate side chains available as binding sites for microorganisms [39], interfering with the biofilm formation process, as well as the quantity and rigidity [40].

The thermopolymerizable acrylic resin (CR) was polished in a horizontal polishing machine with water sandpaper and then in a bench polishing machine with cotton and Spanish white wheels [21]. This was not done for the printed acrylic resins (SP and YL). Because biofilm formation was similar, we can consider that 3D-printed materials do not require this step.

A limitation of this study was the lack of analysis before the characterization of the 3D-printed resins; however, the results can be compared with the literature [2,3,4,10]. Nevertheless, the in vitro evaluation lacks the reproduction of clinical conditions and oral challenges, such as the presence of oral microbiota, pH changes, thermal changes, and masticatory forces. In addition, the specimens used had a flat surface, unlike dentures manufactured for clinical use. Finally, different brands of resin available on the market for the manufacture of complete dentures assume different behaviors according to their composition and printing technology. Thus, future studies should be conducted to eliminate these biases.

## 4. Materials and Methods

The materials used in the study are shown in Table 7.

### 4.1. Specimen Preparation

This laboratory study analyzed two 3D-printed resins in comparison with a conventional heat-polymerized resin using denture hygiene protocols (brushing, brushing plus water, brushing plus sodium hypochlorite at 0.25%, and brushing plus triclosan at 0.15%) for periods of 1, 3, and 5 years. The main variables were color change (ΔE and NBS), surface microhardness, surface roughness (Ra), flexural strength (MPa), and microbial load of complex biofilm composed of *C. albicans*, *S. mutans,* and *S. aureus* formed on resin surfaces, considering longitudinal analysis. Qualitative analyses of the surfaces of the specimens subjected to physical and mechanical tests were performed by confocal microscopy. The sample size was based on previous studies [3,4,40] resulting in 120 specimens for the color change (ΔE and NBS), surface microhardness, and surface roughness, 150 specimens for flexural strength assay, and 114 specimens for microbial load. The flexural strength assay was destructive, as these 10 specimens of each resin were obtained and analyzed at T0.

To evaluate the color stability, hardness, and surface characteristics, a total of 120 circular specimens (12 × 3 mm) were obtained, with dimensions appropriated for the devices used. To evaluate roughness and flexural strength, 150 rectangular specimens (64 × 10 × 3 mm) were obtained, in accordance to ISO 20795-1: 2013 [30], with dimensions as in the literature [41]. To evaluate the microbial load of the biofilm formed on the resin surface, a total of 114 specimens (12 × 3 mm) were obtained.

To obtain the heat-polymerized acrylic resin specimens, metal matrices were inserted into a denture flask (Jon Indústria Brasileira, São Paulo, SP, Brazil) with type III and IV dental stones. Subsequently, the matrices were removed from the flasks, and the molds were filled with heat-polymerized acrylic resin (CR) (Artigos Odontológicos Clássico Ltd.a., São Paulo, SP, Brazil), which was manipulated and polymerized according to the manufacturer’s recommendations. Thus, a 3:1 proportion of powder and liquid was used, and it was polymerized in a 3-h cycle. The finishing of the specimens was performed with a cutter and micromotor. Surfaces were polished in a horizontal polisher (Panambra Industrial e Técnica AS, São Paulo, SP, Brazil) with grit abrasive papers (numbers: 150, 320, 600, and 1200, Norton Saint Gobain Acessórios Ltd.a., Guarulhos, SP, Brazil) and a wet rag wheel with calcium carbonate (Antônio Bussioli ME, Rio Claro, SP, Brazil). The final dimensions of the specimens were confirmed with a pachymeter (Mitutoyo Sul Americana Ltd.a., Suzano, SP, Brazil), and the surface roughness was standardized at 0.2 µm. For the microbiological test, to simulate the polished surface of the prosthesis the roughness of one of the surfaces of the specimen was standardized to a maximum of 0.2 µm. To simulate the internal surface of the prosthesis, the other face of the specimen had the roughness standardized with Ra ranging from 2.7 to 3.7 μm. The roughness was standardized by the Surface Roughness Tester (Mitutoyo Corp, Kawasaki, Japan), with a 0.8 mm cut-off and a 4.8 mm needle stroke.

The 3D-printed specimens were designed and drawn using Rhinoceros 6.0 software (Robert McNeel & Associates, Seattle, WA, USA). For the Yller resin (YL), 3D printing was performed with the Flashforge Hunter 3D Printer (dOne 3D, Ribeirão Preto, SP, Brazil), with a layer height of 0.05 mm parallel to the Z axis. The curing time was 3 s and 20 s for the adhesion layers and 80% light intensity. The printed specimens were washed with ethanol for 3 min, then post-cured for 3 min in a post-curing station (dOne 3D, Ribeirão Preto, SP, Brazil), with a power of 60 W and irradiance of 167.71 mW/cm^2^. Then, they were cleaned with isopropyl alcohol in a washing station (dOne 3D, Ribeirão Preto, SP, Brazil). To print the specimens with SmartPrint (SP) resin, a Miicraft 125 ultra printer (Miicraft, Taiwan, China) was used with a layer height of 0.05 mm parallel to the Z axis and a curing time of 3.3 s. The curing time of the adhesion layers was 30 s at 100% light intensity. Washing was carried out with ethanol for 5 min, followed by a post-cure time of 10 min in a post-cure station (EDG Soluções, São Carlos, SP, Brazil), and cleaning with isopropyl alcohol for 1 min.

To simulate the characterization of the gingiva, one of the surfaces was treated with aluminum oxide (Al_2_O_3_) (60 µm at 4 bar). The specimens were washed and dried, and after 2 min a thin layer of adhesive (Kulzer Mitsui Chemical Group, São Paulo, SP, Brazil) was applied and polymerized for 1 min and 30 s, in a post-curing station (Kulzer Mitsui Chemical Group). Then, a layer of approximately 0.2 mm of R50 (Kulzer Mitsui chemical group) for gingiva was applied and cured for seven minutes. The same was done for the pink pigment (Kulzer Mitsui chemical group). To prevent the formation of the dispersion layer, 0.5 mm thick insulating gel (Kulzer Mitsui chemical group) was applied and cured for fourteen minutes. The quantities of these materials were standardized by weight on scales considering 0.2 mg and 0.5 mg, respectively. After washing, a thin layer of glaze and light-curing sealant (Megadenta Dentalprodukte GmbH, Ribeirão Preto, SP, Brazil) was applied and, after 20 s, cured for 5 min in a polymerizer (Kulzer Mitsui Chemical Group). The entire extrinsic pigmentation process was carried out with clean and fine brushes, always by the same operator.

Each specimen was identified on the side with a marking made by a drill (Labordental Ltda, São Paulo, SP, Brazil).

### 4.2. Hygiene Protocols

The specimens were randomly distributed into four groups (*n* = 10): B—brushing (control 1); W—brushing and immersion in water (control 2); SH—brushing and immersion in 0.25% sodium hypochlorite; and T—brushing and immersion in 0.15% triclosan [25].

Mechanical brushing was performed on a Pepsodent machine (Acess. e Serv. Ltda. ME, Ribeirão Preto, SP, Brazil), according to the ISO 14.569-1 specification (International Organization for Standardization, 2007) [42]. The specimens were brushed with a soft brush (Johnson & Johnson, São José dos Campos, SP, Brazil) using a suspension of neutral soap and distilled water (2 mL:10 mL) [2] at a rate of 356 rpm, 200 g load, and 3.8 cm stroke. The suspension was poured into the machine’s vats over the specimens. Brushing time simulated 12 (T1; 17,800 cycles), 36 (T3; 44,500 cycles), and 60 months (T5; 89,000 cycles) [2]. The suspensions were replaced every 50 min and the brushes every 100 min [27].

To simulate 12, 36, and 60 months of 20 min of daily immersion [28], the specimens were immersed in 200 mL of each solution for 121 h (T1), 242 h (T3 = 363 h), and another 242 h (T5 = 605 h), respectively. The set was kept closed and at room temperature and the solutions were replaced every 24 h.

The variable color changes (ΔE), Knoop microhardness (KHN), surface roughness (µm Ra), and qualitative analysis of the surface were analyzed after obtaining the specimens (T0) and after each period of application of the hygiene protocols. The flexural strength variable was analyzed at T0 and T5. These variables were important to characterize the materials in relation to chewing efforts, variations in the oral environment, and hygiene protocols [1,2,3,5].

### 4.3. Outcomes of Physical and Mechanical Properties 

A portable colorimeter (BYK-Gardner, Geretsried, Germany) was used for color measurements with D65 standardized lighting within the visible spectrum (400 to 700 nm). The Standard Commission Internationale de L’Eclairage (CIE Lab) color system, recommended by the American Dental Association, was used to evaluate the color. This system represents a three-dimensional color space, having components of clarity (*L*), red-green (*a*) and yellow-blue (*b*). The difference in color between the specimens and times can be given using the parameter Δ*Eab*, calculated by the formula: Δ*Eab* = [(Δ*L*)^2^ + (Δ*a*)^2^ + (Δ*b*)^2^] ½. To evaluate the color change relating it to clinical perceptibility, the data were quantified according to the National Bureau of Standards (NBS) units using the following formula: NBS units = ΔE × 0.92; [2]. They were then classified according to: (1) Trace, 0.0–0.5; (2) Slight, 0.5–1.5; (3) Noticeable, 1.5–3.0; (4) Considerable, 3.0–6.0; (5) Very, 6.0–12.0; and (6) Excessive, >12.0.

The measurement of Knoop microhardness was performed with a Shimadzu Microhardness Tester (Shimadzu Corporation, Kyoto, Japan), using a load of 25 g for 5 s. The final measurement was given from the mean of eight readings performed on each specimen.

For the measurement of surface roughness (ΔRa) a rugosimeter (Mitutoyo, Tokyo, Japan) was used, being 4.0 mm in length with a cut-off value of 0.8 mm at a speed of 0.5 mm/s. Three readings were performed on each specimen and the arithmetic mean of three measurements (μm Ra) was calculated.

The flexural strength assessment (EMIC, São Jose dos Pinhais, Paraná, Brazil) was verified at a crosshead speed of 5 mm per minute. Three-point bending tests were carried out using a distance of 50 mm between the two supporting points, and 50 kg was applied to the specimen’s center. Flexural strength was calculated using the formula: *S* = 3*PL*/2*bd*2, where *S* is flexural strength, *P* is the peak load applied, *L* is the span length, *b* is the specimen’s width, and *d* is the specimen’s thickness. The calculation of the maximum flexion of the specimen was achieved from the tension (T) × deformation (d) curve. The results were expressed in kgf/mm^2^ and converted to MPa.

For qualitative analysis of the surface, the specimens were positioned parallel to the table of the confocal microscope Olympus LEXT OLS4000^®^ (Olympus, Tokyo, Honshu, Japan) with the aid of a parallelometer, and 3D images were obtained with a 5× objective and 107 times optical zoom magnification of the most representative areas of each region.

### 4.4. Multispecies Biofilm of C. albicans, S. aureus, and S. mutans

#### 4.4.1. Biofilm Formation

Samples were sterilized by hydrogen peroxide plasma (Advanced Sterilization Products, Irvina, CA, USA). Two additional specimens were immersed in Brain Heart Infusion liquid culture medium (BHI) (Kasvi, São José dos Pinais, PA, Brazil) at 37 °C for 14 days, to confirm the sterilization process due to the absence of microbial growth.

The effectiveness of the hygiene protocols was evaluated (triplicate) against a complex biofilm composed of *Candida albicans* (ATCC 90028; Manassas, VA, USA), *Streptococcus mutans* (ATCC 25175; www.atcc.org, accessed on 15 November 2023), and *Staphylococcus aureus* (ATCC 6538; www.atcc.org, accessed on 15 November 2023). Aseptically, the inoculum of *C. albicans* was cultivated in a liquid culture medium of Sabourand Dextrose Broth, and those of *S. aureus* and *S. mutans* were cultivated in a liquid culture medium of BHI (Kasvi), all at 37 °C for 24 h in a microbiological oven (De Leo—Equipamentos Laboratoriais, Porto Alegre, RS, Brazil), with *S. mutans* maintained under microaerophilic conditions. After centrifuging the suspensions for 5 min at 4200× *g*, the inoculum was standardized in a phosphate buffer saline solution (PBS) and using a spectrophotometer, optical absorbance readings of 0.150 for *S. mutans* and 0.085 to 0.095 for *S. aureus* were given. For *C. albicans*, standardization was performed in a Neubauer chamber. Confirmation of inoculum concentrations was assessed by seeding on agar.

Aseptically, the specimens were distributed in 24-well cell culture plates (Techno Plastic Products, Trasadingen, Switzerland) and each well received 1.5 mL of BHI with the microorganisms at concentrations of 10^6^ for *C. albicans* and 10^7^ for *S. mutans* and *S. aureus*, with the exception of the negative control group, which received sterile culture medium. The plates were incubated at 37 °C for 1 h and 30 min under agitation at 75 rpm (Scientific Equipment, Campinas, SP, Brazil), in microaerophilic conditions. After this period, each specimen and well was washed twice with sterile PBS to remove non-adherent microorganisms. A total of 1.5 mL of sterile culture medium was inserted into each well. The plates were incubated at 37 °C under agitation at 75 rpm for 24 h in microaerophilic conditions and half of the medium was replaced by a new culture medium. The plates were incubated for another 24 h for biofilm maturation.

#### 4.4.2. Hygiene Protocols

Contaminated specimens were randomly distributed into four groups (*n* = 9): NB (no brushing); W (brushing and immersion in water); SH (brushing and immersion in 0.25% sodium hypochlorite); and T (brushing and immersion in 0.15% triclosan) [28]. For NB, specimens were removed from the cell culture plate, rinsed in PBS, and individually inserted into test tubes containing 10 mL of Letheen Broth (LB) medium (HiMedia Laboratories Pvt. Ltd. Mumbai, MH, India).

For W, SH, and T, the specimens were removed from the cell culture plates, washed in PBS, and placed in Plexiglass plates, previously sterilized in a microwave (Consul Facilite, Manaus, AM, Brazil) at 650 W for 6 min, and subsequently brushed for 20 s on each surface of the specimen with a soft brush and mild soap. Brushing was performed manually, by the same operator, with new soft brushes (Johnson & Johnson), sterilized under UV light for 20 min.

After brushing, the specimens were removed from the Plexiglass plates, rinsed in PBS, and individually immersed for 20 min in polypropylene tubes (Techno PlasticProducts-TPP, Trasadingen, Canton Schaffhausen, Switzerland) containing 10 mL of sanitizing solution (water, 0.25% sodium hypochlorite or 0.15% triclosan). The specimens were washed three times in PBS and placed individually in test tubes containing 10 mL of LB medium.

#### 4.4.3. Evaluation of the Microbial Load

The test tube/specimen set was sonicated in ultrasound (Altsonic, Ribeirão Preto, SP, Brazil) at 40 KHz, 200 W for 20 min, and shaken individually in a test tube shaker (Phoenix, Araraquara, SP, Brazil). Then, 0.025 mL of the suspension was seeded in dilutions ranging from 10^0^ to 10^−3^ in Petri dishes containing a specific culture medium. For *C. albicans*, Sabourand Dextrose Ágar (Az Labor, Ribeirão Preto, SP, Brazil) was used; for *S. aureus*, Mannitol Salgado Agar (Interlab Ltd.a., São Paulo, SP, Brazil); and for *S. mutans*, Modified Bacitracin Sucrose Agar (SB-20). Petri dishes were incubated at 37 °C for 48 h in a microbiological oven. For *S. mutans*, incubation was performed in microaerophilic conditions.

CFU counts were performed by calculating CFU/Ml = number of colonies × 10*^n^*/q; where *n* is the absolute value of the dilution, ranging from 0 to 3, and q is the amount in mL of suspension pipetted for dilution, when sowing on the plates, that is, 0.025 [28]. Data were presented in Log_10_CFU/mL. The test tubes with the specimens were incubated at 37 °C for 28 days in a microbiological oven to monitor the presence or absence of turbidity in the culture medium and compare it with the growth of microorganisms in the sown plates.

#### 4.4.4. Data Analysis

The statistical tests were performed using the SPSS 21.0 program (SPSS Inc., Chicago, IL, USA). The Shapiro–Wilk and Levene tests revealed, respectively, normal distribution and homoscedasticity of the color change, hardness, roughness, flexural strength data, and microbial load. Color change, hardness, and roughness data were submitted to the ANOVA Test with repeated measurements, and Tukey’s post-test; the flexural strength data used the ANOVA Test (Three-way) with Tukey’s post-test for all tests. The microbial loads of *C. albicans* and *S. aureus* were submitted to the Generalized Linear Model with Bonferroni Fit. The microbial load of *S. mutans* was submitted to ANOVA Test (Two-Way) with Tukey’s post-test. It was considered α = 0.05.

## 5. Conclusions

Through the results, it can be concluded that:(1)The color variations of the Yller resin after 3 and 5 years, and Smart Print after 5 years, were influenced by immersion in 0.25% sodium hypochlorite;(2)The hardness values of 3D printing resins showed lower compared to the conventional resins, and all protocols promoted a reduction of values in the simulated period of 5 years;(3)The roughness values of 3D printing resins showed higher than the conventional resins, which varied depending on the hygiene protocols;(4)All resins showed a decrease in flexural strength when subjected to all hygiene and control protocols after the 5-year simulation.(5)Brushing associated with immersion in 0.25% sodium hypochlorite was the most efficient protocol, followed by brushing and immersion in 0.15% triclosan. The type of resin did not influence the CFU count, except when the Yller resin was cleaned with brushing and immersion in triclosan and showed a higher count of *C. albicans*.

The results of this study showed that the indication of the dental use of printed resins should not be carried out for prolonged periods, and should be restricted to periods of less than 3 years.

## Figures and Tables

**Figure 1 antibiotics-12-01630-f001:**
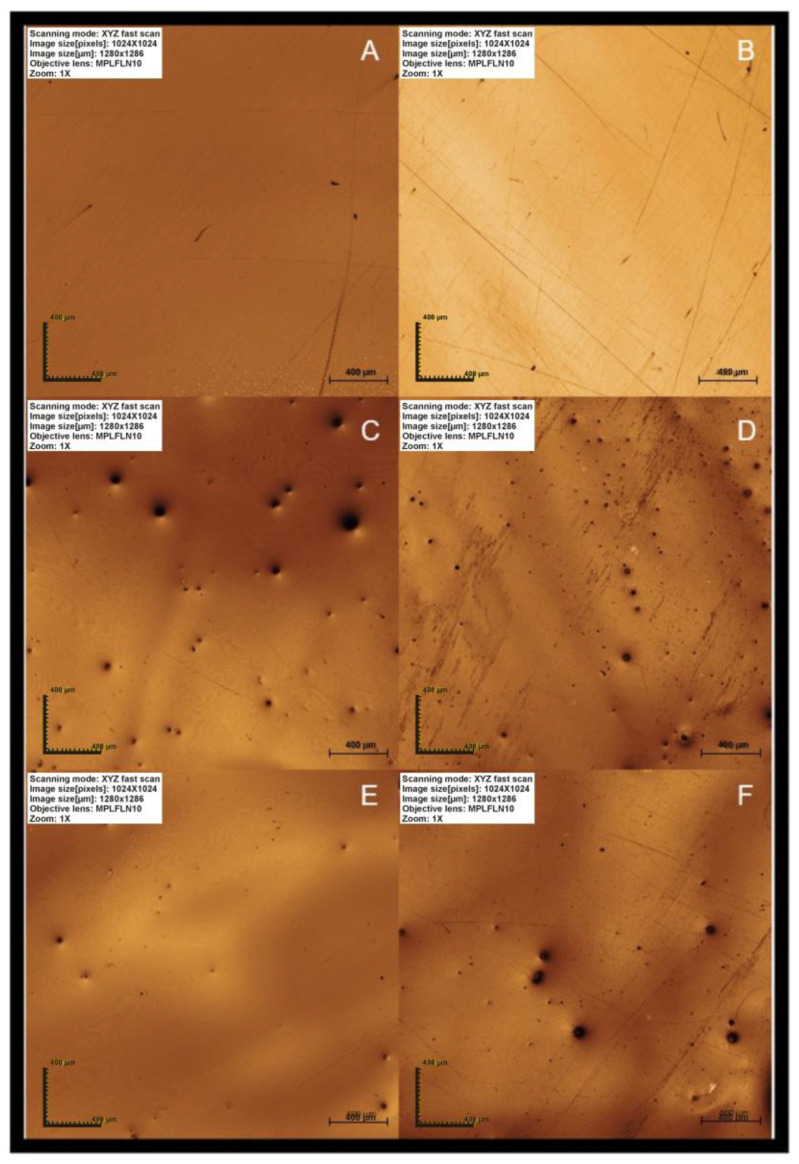
Images of the surface of the resin in baseline (T0) and after 5 years of simulation of the hygiene protocols. (**A**) CR in T0; (**B**) CR after 5 years; (**C**) SP in T0; (**D**) SP after 5 years; (**E**) YL in T0; (**F**) YL after 5 years.

**Table 1 antibiotics-12-01630-t001:** Comparison of the mean (standard deviation) of color change indicated by CIELab (ΔE) and NBS systems.

		T1 (ΔΕ1)			T3 (ΔΕ2)			T5 (ΔΕ3)		
		CR	SP	YL	CR	SP	YL	CR	SP	YL
^#^ CIELab	B	1.28 (0.57) *^Aa^**	1.03 (0.86) *^Aa^**	1.31 (0.56) *^Aa^**	1.19 (0.32) *^Aa^**	0.71 (0.35) *^Aa^**	1.47 (0.53) *^Aa^**	1.38 (0.54) *^Aa^**	0.89 (0.41) *^Aa^**	1.52 (0.5) *^Aa^**
	W	1.36 (0.58) *^Aa^*^◦^*	0.62 (0.32) *^Aa^**	1.45 (0.4) *^Aa^**	1.83 (0.89) *^Aa^**	0.72 (0.32) *^Ba^**	1.87 (0.54) *^Aa^*^◦^*	1.17 (0.61) *^Aa◦^*	0.76 (0.22) *^Aa^**	2.14 (0.62) *^Ba◦^*
	SH	1.5 (0.67) *^Aa^**	0.72 (0.27) *^Ba^**	1.69 (0.36) *^Aa^**	1.53 (0.7) *^ABa^**	0.98 (0.19) *^Aa^**	1.75 (0.62) *^Ba^**	1.27 (0.65) *^Aa^**	1.58 (0.46) *^ABb◦^*	1.93 (0.7) *^Ba^**
	T	1.01 (0.24) *^Aa^*^◦^*	0.49 (0.22) *^Aa^**	1.26 (0.2) *^Aa^**	1.24 (0.44) *^Aa^**	0.73 (0.22) *^Aa^**	1.44 (0.44) *^Aa^**	0.76 (0.18) *^Aa◦^*	0.66 (0.17) *^Aa◦^*	1.56 (0.5) *^Aa^**
** NBS	B	1.18 ^◦^	0.95 ^◦^	1.21 ^◦^	1.09 ^◦^	0.65 ^◦^	1.35 ^◦^	1.27 ^◦^	0.82 ^◦^	1.40 ^◦^
	W	1.25 ^◦^	0.57 ^◦^	1.33 ^◦^	1.68 ^q^	0.66 ^◦^	1.72 ^q^	1.08 ^◦^	0.70 ^◦^	1.97 ^q^
	SH	1.38 ^◦^	0.66 ^◦^	1.55 ^q^	1.41 ^◦^	0.90 ^◦^	1.61 ^q^	1.17	1.45 ^◦^	1.78 ^q^
	T	0.93 ^◦^	0.45 *	1.16 ^◦^	1.14 ^◦^	0.67 ^◦^	1.32 ^◦^	0.7 ^◦^	0.61 ^◦^	1.44 ^◦^

ANOVA test with repeated measurements and Tukey’s post-test. ^#^ Capital letters: compared resins for the same time and hygiene protocol; lowercase letters: compared hygiene protocols for the same resin and time; symbols: compared resins at different times and the same hygiene protocol; equal letters and symbols indicate statistical equality. ** NBS classification: * imperceptible (0.0–0.5); ^◦^ light (0.5–1.5); ^q^ perceptible (1.5–3.0); ^•^ appreciable (3.0–6.0); ^f^ great (6.0–12.0); ^ very large (>12.0). B: brushing, W: brushing and immersion in water, SH: in 0.25% sodium hypochlorite, T: 0.15% triclosan. T1: after 1 year, T3: after 3 years, T5: after 5 years. CR: conventional resin, SP: SmartPrint resin, YL: Yller resin.

**Table 2 antibiotics-12-01630-t002:** Comparison of the mean (standard deviation) of Knoop microhardness (KHN).

	T0			T1			T3			T5		
	CR	SP	YL	CR	SP	YL	CR	SP	YL	CR	SP	YL
B	16.89 (0.84) *^Aa^**	12.31 (1.9) *^Ba^**	10.02 (1.94) *^Aa^**	16.96 (0.28) *^Aa^**	11.72 (1.5) *^Bb^**	11.69 (1.92) *^Ba^**	18.97 (0.6) *^Aa◦^*	9.47 (0.99) *^Ba◦^*	12.52 (2.05) *^Cb^**	18.6 (0.34) *^Ab◦^*	9.87 (1.49) *^Ba◦^*	11.54 (1.84) *^Ca^**
W	17.62 (0.37) *^Aa◦^*	11.95 (1.6) *^Ba^**	9.95 (2.24) *^Cb^**	15.06 (0.26) *^Ab^**	9.62 (1.19) *^Bb◦^*	10.3 (1.42) *^Ba^**	17.05 (0.32) *^Aab◦^*	9.23 (1.29) *^Ba◦^*	11.19 (1.96) *^Cab◦^*	17.35 (0.31) *^Aab◦^*	9.15 (1.07) *^Ba◦^*	9.89 (1.32) *^Bb^**
SH	16.94 (0.35) *^Aa^**	11.03 (2.94) *^Ba◦^*	9.96 (1.62) *^Bb^**	15.42 (0.31) *^Aab◦^*	10.65 (2.12) *^Bab^**	7.43 (1.12) *^Cb◦^*	17.27 (0.61) *^Aab^**	9.39 (1.89) *^Ba◦^*	9.7 (1.97) *^Bb^**	17.45 (0.38) *^Aab^**	9.11 (1.33) *^Ba◦^*	9.72 (1.48) *^Bb^**
T	17.25 (0.22) *^Aa◦^*	11.9 (0.88) *^Ba^**	9.59 (1.35) *^Cb◦^*	14.97 (0.16) *^Ab^**	13.67 (1.85) *^Aa◦^*	11.16 (1.58) *^Ba^**	17.02 (0.2) *^Ab◦^*	11.36 (1.58) *^Bb^**	11.59 (1.87) *^Ba^**	17.02 (0.36) *^Aa◦^*	10.35 (1.46) *^Ba^**	10.43 (1.16) *^Bab^**

ANOVA test with repeated measurements and Tukey’s post-test. Capital letters: compared resins for the same time and hygiene protocol; lowercase letters: compared hygiene protocols for the same resin and time; symbols (* and *◦*): compared resins at different times and the same hygiene protocol; equal letters and symbols indicate statistical equality. B: brushing, W: brushing and immersion in water, SH: in 0.25% sodium hypochlorite, T: 0.15% triclosan. T0: baseline, T1: after 1 year, T3: after 3 years, T5: after 5 years. CR: conventional resin, SP: SmartPrint resin, YL: Yller resin.

**Table 3 antibiotics-12-01630-t003:** Comparison of the means (standard deviations) of the surface roughness (Ra, μm) by the interaction between resins × hygiene protocols and resins × times.

	* Interaction Resins × Hygiene Protocols				** Interaction Resins × Times			
	B	W	SH	T	T0	T1	T3	T5
CR	0.05 (0.008) *^Aa^*	0.05 (0.01) *^Aa^*	0.06 (0.01) *^Aa^*	0.05 (0.01) *^Aa^*	0.05 (0.01) *^Aa^*	0.05 (0.01) *^Aa^*	0.05 (0.01) *^Aa^*	0.05 (0.01) *^Aa^*
SP	0.65 (0.13) *^Ab^*	0.82 (0.29) *^Ab^*	0.95 (0.25) *^Bb^*	0.92 (0.15) *^Bb^*	0.90 (0.27) *^Ab^*	0.85 (0.25) *^Ab^*	0.82 (0.23) *^ABb^*	0.80 (0.20) *^Bb^*
YL	1.36 (0.40) *^Ac^*	1.32 (0.31) *^Ac^*	1.16 (0.25) *^ABb^*	0.94 (0.30) *^Bb^*	1.20 (0.31) *^Ac^*	1.20 (0.40) *^Ac^*	1.18 (0.40) *^Ac^*	1.21 (0.33) *^Ac^*

* ANOVA test with repeated measurements and Tukey’s post-test. Capital letters: compared hygiene protocols for the same resin; lowercase letters: compared resins for the same hygiene protocols; ** capital letters: compared times for the same resin; lowercase letters: compared resins for the same time; equal letters and symbols indicate statistical equality. B: brushing, W: brushing and immersion in water, SH: in 0.25% sodium hypochlorite, T: brushing and immersion 0.15% triclosan. T0: baseline, T1: after 1 year, T3: after 3 years, T5: after 5 years. CR: conventional resin, SP: SmartPrint resin, YL: Yller resin.

**Table 4 antibiotics-12-01630-t004:** Comparison of the means (standard deviations) of the flexural strength (MPa) by interaction between time × hygiene protocols.

	* T0	T5
B	83.27 (9.73) *^Aa^*	76.30 (12.74) *^Ba^*
W	83.27 (9.73) *^Aa^*	75.20 (8.41) *^Ba^*
SH	83.27 (9.73) *^Aa^*	67.08 (7.30) *^Bb^*
T	83.27 (9.73) *^Aa^*	67.90 (10.90) *^Bb^*

ANOVA test (three-way) with Tukey’s post-test. * The data obtained in T0 was used for comparison with data in T5 in the groups of hygiene protocols; capital letters: compared times for the same hygiene protocols; lowercase letters: compared hygiene protocols for the same time; equal letters and symbols indicate statistical equality. B: brushing, W: brushing and immersion in water, SH: brushing and immersion in 0.25% sodium hypochlorite, T: brushing and immersion of 0.15% triclosan. T0: baseline, T5: after 5 years.

**Table 5 antibiotics-12-01630-t005:** Comparison of the interaction between the CFU count factors (log10 + 1) of *C. albicans*, after the use of different hygiene protocols.

		NB	W	T	*p*
CR	Mean (SD)	5.09 (0.47)	3.20 (0.38)	1.59 (1.06)	0.033
	Median	4.95 *^Aa^*	3.20 *^Ca^*	1.61 *^Ba^*	
	CI	4.73–5.46	2.90–3.49	0.77–2.41	
SP	Mean (SD)	5.72 (0.75)	4.06 (0.71)	1.88 (1.46)	
	Median	5.89 *^Aa^*	4.44 *^Ca^*	2.72*^Bab^*	
	CI	5.14–6.30	3.51–4.60	0.75–3.01	
YL	Mean (SD)	5.89 (0.76)	2.95 (1.78)	2.82 (0.65)	
	Median	5.98 *^Aa^*	3.25 *^Ba^*	3.03 *^Bb^*	
	CI	5.30–6.48	1.58–4.32	2.31–3.32	

Generalized linear model with Bonferroni post-test. Equal letters indicate statistical equality; capital letters: a comparison between protocols for the same resin; lowercase letters: a comparison between resins from the same protocol. SD: standard deviation; CI: confidence interval. Comparison between resins: *p* = 0.041. Comparison between protocols: *p* < 0.001. NB: no brushing; W: brushing and immersion in water; T: brushing and immersion 0.15% triclosan.

**Table 6 antibiotics-12-01630-t006:** Comparison of CFU counts (log10 + 1) of *S. aureus*, after using different hygiene protocols.

	NB	W	T
Mean (SD)	7.42 (0.35)	3.96 (0.38)	0.58 (1.19)
Median	7.5 *^A^*	4.0 *^B^*	0 *^C^*
CI	7.13–7.70	3.67–4.24	0.29–0.86

Generalized linear model with Bonferroni post-test. Equal letters indicate statistical equality; capital letters: comparison between protocols. SD: standard deviation; CI: confidence interval. Comparison between resins: *p* = 0.363; interaction: *p* = 0.374. NB: no brushing. W: brushing and immersion in water; T: brushing and immersion 0.15% triclosan.

**Table 7 antibiotics-12-01630-t007:** Materials used in the study.

Brand Name	Manufacturer	Batch Numbers
Heat-polymerized acrylic resin medium pink color	Clássico Artigos Odontológicos, Campo Limpo Paulista, SP, Brazil	050514
Resin Yller Cosmos Denture medium pink color	Yller Biomaterials, Pelotas, RG, Brazil	-
Resin SmartPrint Bio Denture medium pink color	SmartDent, São Carlos, SP, Brazil	1547
Signum connector	Kulzer Mitsui Chemical Group, São Paulo, SP, Brazil	K010518
R50 gingiva flow, Pala Cre active	Kulzer Mitsui Chemical Group, São Paulo, SP, Brazil	K010518
Colorfluid pink, Pala Cre active	Kulzer Mitsui Chemical Group, São Paulo, SP, Brazil	K010126
Signum insulating gel	Kulzer Mitsui Chemical Group, São Paulo, SP, Brazil	K010128
Megaseal	Megadenta Dentalprodukte GmbH, Ribeirão Preto, SP, Brazil	4G044A
Toothbrush Tek soft bristles	Johnson & Johnson do Brasil Ind. e Com. Prod. para Saúde Ltda., S. J. dos Campos, SP, Brazil	540557
Neutral soap—sodium lauryl sulfate, diethanolamine, cocamidopropyl, betaine, methylparaben, polyquatemium 7, citric acid, polyethylene glycol, pearl base, perfume and water	Pleasant, Perol Com. e Ind. Ltda., Ribeirão Preto, São Paulo, Brazil	110127
* Triclosan (** 10 mL of 0.056M sodium hydroxide solution + 0.15 g Triclosan = 0.15% (1.5 mg/mL)	* Mix das essências, Belo Horizonte, MG, Brazil.	107M4876V
Sodium hypochlorite (** 0.25% sodium hypochlorite)	Super Candida^®^ Indústria Anhembi, Osasco, SP, Brazil	E67144212MD2

* Pure active principle; ** Solutions prepared at the Oral Rehabilitation Research Laboratory at the College of Dentistry of Ribeirão Preto.

## Data Availability

The data presented in this study are available on request from the corresponding author.

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
