# Peer review of "Effect of Hygiene Protocols on the Mechanical and Physical Properties of Two 3D-Printed Denture Resins Characterized by Extrinsic Pigmentation as Well as the Mixed Biofilm Formed on the Surface"

_antibiotics, 2023, doi:10.3390/antibiotics12111630_

Round 1

Reviewer 1 Report

Comments and Suggestions for Authors

Very interesting article, 3D materials are new on the market, so testing them is highly desirable. A lot of work well done. Sample preparation, microorganism cultivation, disinfection were taken care of. I am full of admiration for your work.

  As a reviewer, I have a few minor comments on the text:

Introduction

composed of PMMA associated with two or more polymerizable groups - PMMA is a polyester derived from methyl methacrylate, which has only one C=C group (which can polymerized by free radical polymerization process), which means it can be monofunctional. 3D printing materials use other methacrylate compounds similar to those contained in composite materials, e.g. UDMA, bis EMA, and acrylates. . Please correct it, thank you.

However, there are  few studies that present results on the behavior of pigmented base resins- Changing the color of 3D printing resins is described, for example, in the article, you can see how the surface of such resin can change  during time

Raszewski Z, Chojnacka K, Mikulewicz M. Effects of Surface Preparation Methods on the Color Stability of 3D-Printed Dental Restorations. J Funct Biomater. 2023 May 5;14(5):257. doi: 10.3390/jfb14050257. PMID: 37233367; PMCID: PMC10219081.

Dimitrova, M.; Chuchulska, B.; Zlatev, S.; Kazakova, R. Colour Stability of 3D-Printed and Prefabricated Denture Teeth after Immersion in Different Colouring Agents—An In Vitro Study. Polymers 202214, 3125. https://doi.org/10.3390/polym14153125

Materials and methods

Table 7 It would be good to add a little more information about the tested products, e.g. composition of 3D printing resins, colors, batch numbers.

The main variables were color change (ΔE and NBS) - it would be good to explain what the color change scale means, it is the leather you use so it would be good to explain it, thank you

To evaluate the color stability, hardness, and surface characteristics, a total of 120

circular specimens (12 x 3 mm) were obtained; to evaluate the roughness and flexural

strength, a total of 150 rectangular specimens (65 x 10 x 3.3 mm) were obtained- why did you choose such sample dimensions? e.g. FS is given in the ISO Denture Base Polymers standard

and the molds  were filled with heat-polymerized acrylic resin (CR) (Clássico; Artigos Odontológicos- It would be good to provide some proportions of the resin mixture because it affects many of the parameters you measure. The more powder and less liquid, the higher the FS and hardness. Polymerization time also matters, as you know.

post-cured for  three mins (Forno de Cura, dOne 3D, Ribeirão Preto, São Paulo, Brazil), and cleaned with isopropyl alcohol (Forno de Cura, dOne 3D, Ribeirão Preto, São Paulo, Brazil- For hardening, you used a light furnace with what power? But why did you use this device to wash it with alcohol? I would expect an ultrasonic cleaner or something similar?

Megadenta Dentalprodukte GmbH, Ribeirão Preto, São Paulo, Brazil) was applied and, after twenty sec, cured for five minutes- And what kind of light furnace was it hardened in?

the ISO 14.569-1 specification (International Organization for Standardization, 1999-  there is new version of  this standard- ISO 14569-1:2007 - Dental materials and without ,,.’’

The equations could be written using the equation editor. Then give a space and then explain what the individual variables in the equation mean, but this is only from a graphical and optical point of view.

   However, it seems to me that it would be good to make a graphic abstract of your actions. It will be more readable and understandable at first glance, because you have really done a lot of work. I admire that.

Paraná, Brazil) at 37ºC for 14 days, to confirm the sterilization process due to thethe absence of microbial grow- remove on ,,the’’

biofilm composed of Candida albicans (ATCC 90028), Streptococcus mutans (ATCC) producer country?

placed in Plexiglass plates, previously sterilized in a microwave (Consul Fa[1]cilite, Manaus, AM, Brazil) at 650W for 60 min- 60 minutes of sterilization did PMMA not dissolve in these conditions?

Results

Table 4 You have exceptionally low MPa values for FS, the standard says that the pressure on denture plates should be above 60 MPa. 12 MPa is a very brittle material that does not meet the standard and is likely to break immediately during use.

Discussion

 This is from results:

Color change was influenced by the interaction among protocol × material × time (p

= 0.036). The greatest color changes (ΔE) were observed with the Yller (YL) resin after 3

and 5 years of exposure to the sodium hypochlorite (SH)

and this is from Discussion: The results indicated that the protocol using immersion in sodium hypochlorite at 0.25% generated a significant color change of the resins after the simulation of 1, 3 and 5 years.- So my question is, did you conduct research calculating the effects of aging tests or were these tests conducted over 5 years?

inferior hardness values compared to the heat-polymerized resin. This result is in accordance with other studies.- You write that 3D printing resins have a lower FS than PMMA resin, but are harder? If they have a smaller amount of unpolymerized bonds, should they also have lower surface hardness?

Conclusion

good conclusion, but I would change the beginning so that each sentence does not start with The printed resin - e.g. dentures made in 3D printing, printed samples, etc.

References

23. International Standards Organization. ISO 1567:1999. Dentistry - denture base polymers.- new version ISO 20795-1:2013 - Dentistry — Base polymers — Part 1

good luck in further research!

Author Response

Response Letter

Manuscript ID: antibiotics-2693251: Effect of hygiene protocols on the mechanical and physical properties of two 3D-printed denture resins characterized by extrinsic pigmentation as well as the mixed biofilm formed on the surface.

Submission to Antibiotics.

Dear Editor and Reviewers,

We want to thank you for your attention in revising our work, and we appreciate the given comments and suggestions.

We elaborated this letter as a point-by-point response for the reviewers. The suggestions were considered, and our answers to each were written below in blue. The alterations or additions to the manuscript's text were written with the yellow highlight.

We hope to have reached your expectations.

Respectfully,

The authors.

___________________________________________________________________________

Reviewer #1

Introduction

Comments/Suggestion: composed of PMMA associated with two or more polymerizable groups - PMMA is a polyester derived from methyl methacrylate, which has only one C=C group (which can polymerized by free radical polymerization process), which means it can be monofunctional. 3D printing materials use other methacrylate compounds similar to those contained in composite materials, e.g. UDMA, bis EMA, and acrylates. . Please correct it, thank you.

Answer: We thank you for the note. In the manuscript, the information was updated, adding a new reference that correctly names 3D printing resins, as "methacrylate based photocurable 3D resin". 

Text Change: Aiming at reducing the number of clinical visits, predicting results, simplifying laboratory work, and improving the quality of denture devices, Computer-Aided Manufacturing (CAD/CAM) can be used for complete denture manufacturing [2–4], either through the subtractive technique performed by milling processed PMMA blocks, which is a polyester derived from methyl methacrylate with only one C=C group and can be polymerized by free radical polymerization process or, by 3D printing materials, one photosensitive resin composed by a photosensitive prepolymer, reactive monomer, photoinitiator, and other additives.  [4–9].

Comments/Suggestion: However, there are  few studies that present results on the behavior of pigmented base resins- Changing the color of 3D printing resins is described, for example, in the articleyou can see how the surface of such resin can change  during time.

Raszewski Z, Chojnacka K, Mikulewicz M. Effects of Surface Preparation Methods on the Color Stability of 3D-Printed Dental Restorations. J Funct Biomater. 2023 May 5;14(5):257. doi: 10.3390/jfb14050257. PMID: 37233367; PMCID: PMC10219081.

Dimitrova, M.; Chuchulska, B.; Zlatev, S.; Kazakova, R. Colour Stability of 3D-Printed and Prefabricated Denture Teeth after Immersion in Different Colouring Agents—An In Vitro Study. Polymers 202214, 3125. https://doi.org/10.3390/polym14153125

Answer: In the manuscript, the two references were inserted and we updated the sentence according to what is in the literature.

Text Change: Some studies present results on the behavior of printed resins after exposure to pigmented solutions, such as coffee and red wine [11, 12]. However, literature is scarce regarding the behavior of base resins characterized and exposed to long periods of hygiene protocols [13, 14].

Materials and methods

Comments/Suggestion: Table 7 It would be good to add a little more information about the tested products, e.g. composition of 3D printing resins, colors, and batch numbers.

Answer: In table 7, the color of the resins and the batch numbers used were added, except for the Yller resin, since the finished specimens were provided by the manufacturer itself, we did not have access to information about the batch.

Text Change:

Brand Name

Manufacturer

Batch Numbers

Heat-polymerized acrylic resin       medium pink color

Clássico Artigos Odontológicos, Campo Limpo Paulista, SP, Brazil

050514

Resin Yller Cosmos Denture           medium pink color

Yller Biomaterials, Pelotas, RG, Brazil

-

Resin SmartPrint Bio Denture        medium pink color

SmartDent, São Carlos, SP, Brazil

1547

Signum connector

Kulzer Mitsui chemical group, São Paulo, SP, Brazil

K010518

R50 gingiva flow, Pala Cre active

Kulzer Mitsui chemical group, São Paulo, SP, Brazil

K010518

Colorfluid pink, Pala Cre active

Kulzer Mitsui chemical group, São Paulo, SP, Brazil

K010126

Signum insulating gel

Kulzer Mitsui chemical group, São Paulo, SP, Brazil

K010128

Megaseal

Megadenta Dentalprodukte GmbH, Ribeirão Preto, SP, Brazil

4G044A

Toothbrush Tek soft bristles

Johnson & Johnson do Brasil Ind. e Com. Prod. para Saúde Ltda., S. J. dos Campos, SP, Brazil

540557

Neutral soap - sodium lauryl sulfate, diethanolamine, cocamidopropyl, betaine, methylparaben, polyquatemium 7, citric acid, polyethylene glycol, pearl base, perfume and water

Pleasant, Perol Com. e Ind. Ltda., Ribeirão Preto, São Paulo, Brazil

110127

*Triclosan (**10 mL of 0.056M sodium hydroxide solution + 0.15g Triclosan = 0.15% (1.5mg/mL)

*Mix das essências, Belo Horizonte, MG, Brazil.

107M4876V

Sodium hypochlorite (** 0.25% sodium hypochlorite)

Super Candida ® Indústria Anhembi, Osasco, SP, Brazil

E67144212MD2

*Pure active principle; ** Solutions prepared at the Oral Rehabilitation Research Laboratory at the College of Dentistry of Ribeirão Preto.

Comments/Suggestion: The main variables were color change (ΔE and NBS) - it would be good to explain what the color change scale means, it is the leather you use so it would be good to explain it, thank you

Answer: In the manuscript, descriptions of the Cie-Lab and NBS systems were added. Thanks for the note.

Text Change: The Standard Commission Internationale de L’Eclairage (CIE Lab) color system, recommended by the American Dental Association, was used to evaluate the color. This system represents a three-dimensional color space, having components of clarity (L), red-green (a) and yellow-blue (b). The difference in color between the specimens and times can be given using the parameter ΔEab, calculated by the formula: Δ Eab = [(ΔL)² + (Δa)² + (Δb)²] ½ . To evaluate the color change relating it to clinical perceptibility, the data were quantified according to the National Bureau of Standards (NBS)…

Comments/Suggestion: To evaluate the color stability, hardness, and surface characteristics, a total of 120 circular specimens (12 x 3 mm) were obtained; to evaluate the roughness and flexural strength, a total of 150 rectangular specimens (65 x 10 x 3.3 mm) were obtained- why did you choose such sample dimensions? e.g. FS is given in the ISO Denture Base Polymers standard.

Answer: The dimensions of the specimens for color and hardness tests were defined in order to facilitate the use of the necessary equipment (portable spectrophotometer and microhardness meter, respectively). Thus, as the spectrophotometer requires a minimum diameter of the specimen to carry out color reading without external interference, we manufacture the specimens in order to enable the equipment to be used safely. Furthermore, as we carried out 4 microhardness measurements per specimen, a minimum specimen size would be necessary for all assessments to be carried out. For the flexion and roughness test, we used dimensions established in ISO.

Text Change: To evaluate the color stability, hardness, and surface characteristics, a total of 120 circular specimens (12 x 3 mm) were obtained, with dimensions appropriated  for e devices used; to evaluate the roughness and flexural strength 150 rectangular specimens (64 x 10 x 3 mm) were obtained in accordance with ISO 20795-1: 2013 with dimensions to the literature [30];  to evaluate the microbial load of biofilm formed on resin surface, a total of 114 specimens (12 x 3 mm) were obtained.

Comments/Suggestion: and the molds  were filled with heat-polymerized acrylic resin (CR) (Clássico; Artigos Odontológicos- It would be good to provide some proportions of the resin mixture because it affects many of the parameters you measure. The more powder and less liquid, the higher the FS and hardness. Polymerization time also matters, as you know.

Answer: To handle the acrylic resin, the proportion indicated by the manufacturer was used, 3:1 of powder and liquid, respectively, and it was polymerized in a 3-hour cycle, reaching a maximum temperature of 94°C (201.2 F).

Text Change: Thus, a 3:1 proportion of powder and liquid was used, and it was polymerized in a 3-hour cycle.

Comments/Suggestion: post-cured for  three mins (Forno de Cura, dOne 3D, Ribeirão Preto, São Paulo, Brazil), and cleaned with isopropyl alcohol (Forno de Cura, dOne 3D, Ribeirão Preto, São Paulo, Brazil- For hardening, you used a light furnace with what power? But why did you use this device to wash it with alcohol? I would expect an ultrasonic cleaner or something similar?

Answer: For washing and post-curing the specimens, the Ciclone 3D post-printing washing and curing station (Done 3D, Ribeirão Preto, São Paulo, Brazil) was used. It performs both procedures in a single piece of equipment, reducing the pace of work. For washing, the equipment operates with a propeller, which agitates the alcohol, optimizing the cleaning of the species. For post-curing, it operates with UV light with a nominal power of 60W and an irradiance of 167.71 mW/cm2.

Text Change: The printed specimens were washed with ethanol for three minutes, post-cured for three minutes in a post-curing station (dOne 3D, Ribeirão Preto, São Paulo, Brazil) with a power of 60 W and irradiance of 167.71 mW/cm2. Then, they were cleaned with isopropyl alcohol in a washing station (dOne 3D, Ribeirão Preto, São Paulo, Brazil).

Comments/Suggestion: Megadenta Dentalprodukte GmbH, Ribeirão Preto, São Paulo, Brazil) was applied and, after twenty sec, cured for five minutes- And what kind of light furnace was it hardened in?

Answer: The UniXS polymerizer (Kulzer Mitsui Chemical Group, São Paulo, São Paulo, Brazil) was used to polymerize the pigments.

Text Change: The quantities of these materials were standardized by weight on scales considering 0.2 mg and 0.5 mg, respectively. After washing, a thin layer of glaze and Megaseal  light-curing sealant (Megadenta Dentalprodukte GmbH, Ribeirão Preto, São Paulo, Brazil) was applied and, after twenty sec, cured for five minutes in a polymerizer (Kulzer Mitsui Chemical Group).

Comments/Suggestion: the ISO 14.569-1 specification (International Organization for Standardization, 1999-  there is new version of  this standard- ISO 14569-1:2007 - Dental materials and without ,,.’’

Answer: Thanks for pointing out this observation. We have updated it to the latest version.

Text Change:

-… according to the according to the ISO 14.569-1 specification (International Organization for Standardization, 2007) [42].

- International Organization for Standardization, 2007. Technical Specification 14569-1. Dental Materials –Guidance on testing of wear resistance – Part 1: Wear by tooth brushing. Switzerland, ISO.

Comments/Suggestion: The equations could be written using the equation editor. Then give a space and then explain what the individual variables in the equation mean, but this is only from a graphical and optical point of view.

Answer: Thanks for your observation. The equations were changed and legends were added.

Text Change: The Standard Commission Internationale de L’Eclairage (CIE Lab) color system, recommended by the American Dental Association, was used to evaluate the color. This system represents a three-dimensional color space, having components of clarity (L), red-green (a) and yellow-blue (b). The difference in color between the specimens and times can be given using the parameter ΔEab, calculated by the formula: Δ Eab = [(ΔL)² + (Δa)² + (Δb)²] ½ .

S = 3PL / 2bd2, where S is flexural strength, P is the peak load applied, L is the span length, b is the specimen’s width, and d is the specimen’s thickness.

Comments/Suggestion: However, it seems to me that it would be good to make a graphic abstract of your actions. It will be more readable and understandable at first glance, because you have really done a lot of work. I admire that.

Answer: Thank you for pointing out this observation. The grafic abstract were made in the abstract section, conformed suggested.

Comments/Suggestion: Paraná, Brazil) at 37ºC for 14 days, to confirm the sterilization process due to thethe absence of microbial grow- remove on ,,the’’

Answer: Thank you for pointing out this observation. The preposition were removed, conformed suggested.

Text Change: 37ºC for 14 days, to confirm the sterilization process due to the absence of microbial growth.

Comments/Suggestion: biofilm composed of Candida albicans (ATCC 90028), Streptococcus mutans (ATCC) producer country?

Answer: Thank you very much for the suggestion. When carrying out a consultation on the ATCC website (https://www.atcc.org), we found that Candida albicans (ATCC 90028) contains the information on geographic isolation: United States; Iowa. However, when consulting the strains of Streptococcus mutans (ATCC 25175) and Staphylococcus aureus (ATCC 6538), unfortunately we did not find this information available.

Text Change: The effectiveness of the hygiene protocols was evaluated (triplicate) against a complex biofilm composed of Candida albicans (ATCC 90028; United States, Iowa), Streptococcus mutans (ATCC 25175; www.atcc.org), and Staphylococcus aureus (ATCC 6538; www.atcc.org).

Comments/Suggestion: placed in Plexiglass plates, previously sterilized in a microwave (Consul Fa[1]cilite, Manaus, AM, Brazil) at 650W for 60 min- 60 minutes of sterilization did PMMA not dissolve in these conditions?

Answer: Thank you for pointing out this observation. There was probably a typing error, sterilization lasted 6 minutes, the change with the correct information was inserted.

Text Change:… microwave (Consul Facilite, Manaus, AM, Brazil) at 650W for 6 minutes, and subsequently brushed for 20 seconds on each surface of the specimen with a soft brush and mild soap.

Results

Comments/Suggestion: Table 4 You have exceptionally low MPa values for FS, the standard says that the pressure on denture plates should be above 60 MPa. 12 MPa is a very brittle material that does not meet the standard and is likely to break immediately during use.

Answer: Thank you for pointing out this observation. The results were show in kgf and not MPa. We apologize for this. The values in Kgf were substituted by values in MPa.

Text Change:

*T0

T5

B

83.27 (9.73)

Aa

76.30 (12.74)

Ba

W

83.27 (9.73)

Aa

75.20 (8.41)

Ba

SH

83.27 (9.73)

Aa

67.08 (7.30)

Bb

T

83.27 (9.73)

Aa

67.90 (10.90)

Bb

Discussion

Comments/Suggestion: This is from results: Color change was influenced by the interaction among protocol × material × time (p = 0.036). The greatest color changes (ΔE) were observed with the Yller (YL) resin after 3 and 5 years of exposure to the sodium hypochlorite (SH)

and this is from Discussion: The results indicated that the protocol using immersion in sodium hypochlorite at 0.25% generated a significant color change of the resins after the simulation of 1, 3 and 5 years.- So my question is, did you conduct research calculating the effects of aging tests or were these tests conducted over 5 years?

Answer: Thank you for pointing out this observation. Changes to this section of the results were made to clarify that the effects of aging tests were simulated.

Text Change: Color change was influenced by the interaction among protocol × material × time (p = 0.036). The greatest color changes (ΔE) were observed with the Yller (YL) resin after the simulation of 3 and 5 years of exposure to the sodium hypochlorite (SH) protocol when compared to the other resins.

Comments/Suggestion: inferior hardness values compared to the heat-polymerized resin. This result is in accordance with other studies.- You write that 3D printing resins have a lower FS than PMMA resin, but are harder? If they have a smaller amount of unpolymerized bonds, should they also have lower surface hardness?

Answer: Thank you for pointing out this observation. We have redacted this section again to clarify that the reduction in hardness in printed resins was bigger than in thermopolymerized resins.

Text Change: It is worth mentioning that the reduction in hardness of the printed resin was bigger than that of the heat-polymerized resin.

Conclusion

Comments/Suggestion: good conclusion, but I would change the beginning so that each sentence does not start with The printed resin - e.g. dentures made in 3D printing, printed samples, etc.

Answer: Thank you for the suggestion. The beginning of sentences have been changed.

Text Change:

  • The color variation of the Yller resin after 3 and 5 years, and Smart Print after 5 years was influenced by immersion in 0.25% sodium hypochlorite;
  • The hardness values of 3D printing resins showed lower ​​compared to the conventional resins and all protocols promoted a reduction of values ​​in the simulated period of 5 years;
  • The roughness values of 3D printing resins showed higher ​​than the conventional resins, which varied depending on the hygiene protocols;

References

Comments/Suggestion: 23. International Standards Organization. ISO 1567:1999. Dentistry - denture base polymers.- new version ISO 20795-1:2013 - Dentistry — Base polymers — Part 1.

Answer: Thanks for pointing out this observation. We have updated it to the latest version in the references.

Text Change:

- Despite the decrease found, the values ​​respect the recommendations by the ISO standard 20795-1 (65 Mpa) [29].

- 30. International Organization for Standardization, 2013. Technical Specification 20795-1. Dentistry - Base polymers - Part 1: denture base polymers. 2nd ed. Switzerland, ISO.

Reviewer 2 Report

Comments and Suggestions for Authors

None

Author Response

Dear Reviewer,

All suggestions were considered and responded to one by one in the PDF file and the changes can be found in the manuscript file.

Reviewer 3 Report

Comments and Suggestions for Authors

This manuscript presents a study on the effect of hygiene protocols on the physical-mechanical properties and colony forming units of Candida albicans, Staphylococcus aureus, and Streptococcus mutans on two 3D-printed denture resins with extrinsic pigmentation compared to conventional resin. Generally, the data support the result. However, the author should recheck the whole manuscript to avoid some writing mistakes.

The imagines in  Figure 1 lack scale bar. In the legend, please revise “CR in T0” to “A) CR in T0”.

Ensure that abbreviations are expanded upon their first use. For example, in the first three lines of page 2, the abbreviation "3D" is used for "three-dimensional". You've expanded this twice, which is unnecessary. This issue also appears in the third paragraph of the "3. Discussion" section.

In the "4. Materials and Methods" section, the details provided for the materials are inconsistent. Some entries mention the manufacturer and country, while others list the brand and catalog number. Please ensure uniformity in this section.

Comments on the Quality of English Language

Minor editing of English language required

Author Response

Response Letter

Manuscript ID: antibiotics-2693251: Effect of hygiene protocols on the mechanical and physical properties of two 3D-printed denture resins characterized by extrinsic pigmentation as well as the mixed biofilm formed on the surface.

Submission to Antibiotics.

Dear Editor and Reviewers,

We want to thank you for your attention in revising our work, and we appreciate the given comments and suggestions.

We elaborated this letter as a point-by-point response for the reviewers. The suggestions were considered, and our answers to each were written below in blue. The alterations or additions to the manuscript's text were written with the yellow highlight.

We hope to have reached your expectations.

Respectfully,

The authors.

___________________________________________________________________________

Reviewer #3

Comments/Suggestion: This manuscript presents a study on the effect of hygiene protocols on the physical-mechanical properties and colony forming units of Candida albicans, Staphylococcus aureus, and Streptococcus mutans on two 3D-printed denture resins with extrinsic pigmentation compared to conventional resin. Generally, the data supports the result. However, the author should recheck the entire manuscript to avoid some writing mistakes"

Answer: We read the manuscript carefully and corrected any writing errors.

Comments/Suggestion: The imagines in Figure 1 lack scale bar. In the legend, please revise “CR in T0” to “A) CR in T0.

Answer: The scale bar was inserted in the image and the caption, we corrected the information regarding the hygiene protocol.

Text Change: Images of the surface of the resin in baseline (T0) and after 5 years of simulation of the hygiene protocols. A) CR in T0; B) CR after 5 years; C) SP in T0; D) SP after 5 years; E) YL in T0; F) YL after 5 years.

Comments/Suggestion: “Ensure that abbreviations are expanded upon their first use. For example, in the first three lines of page 2, the abbreviation "3D" is used for "three-dimensional". You've expanded this twice, which is unnecessary. This issue also appears in the third paragraph of the "3. Discussion" section”.

Answer: Thanks for your observation. We corrected the expansions of the term "Three-dimensional", keeping its expansion only the first time, it appears in the text.

Text change:

Introduction: 3D printing represents a method of lower-cost processing [10].

Discussion: The study emphasized the digital technologies of computer-aided design/computer-aided manufacturing (CAD/CAM) and 3D printing

Comments/Suggestion: “In the "4. Materials and Methods" section, the details provided for the materials are inconsistent. Some entries mention the manufacturer and country, while others list the brand and catalog number. Please ensure uniformity in this section”.

Answer: Thanks for your observation. In section 4. Materials and Methods, references to the products used were standardized, keeping only the manufacturer, as well as their city, state and country.

Text Change:

-To obtain the heat-polymerized acrylic resin specimens, metal matrices were inserted into a denture flask (Jon Indústria Brasileira, São Paulo, São Paulo, Brazil).

-…and the molds were filled with heat-polymerized acrylic resin (CR) (Artigos Odontológicos Clássico Ltda., São Paulo, São Paulo, Brazil)

-...were polished in a horizontal polisher (Panambra Industrial e Técnica AS, São Paulo, São Paulo, Brazil)

-...and a wet rag wheel with calcium carbonate (Antônio Bussioli ME, Rio Claro, São Paulo, Brazil).

-The final dimensions of the specimens were confirmed with a pachymeter (Mitutoyo Sul Americana Ltda., Suzano, São Paulo, Brazil)

-The roughness was standardized by the Surface Roughness Tester (Mitutoyo Corp, Kawasaki, Japan)

-The 3D-printed specimens were designed and drawn using Rhinoceros 6.0 software (Robert McNeel & Associates, Seattle, Washington, USA).

-For the Yller resin (YL), 3D printing was performed with the Flashforge Hunter 3D Printer (dOne 3D, Ribeirão Preto, São Paulo, Brazil)

-The printed specimens were washed with ethanol for three minutes, post-cured for three minutes in a washing and post-curing station (dOne 3D, Ribeirão Preto, São Paulo, Brazil).

-They were cleaned with isopropyl alcohol, in a washing and post-curing station (dOne 3D, Ribeirão Preto, São Paulo, Brazil).

-Washing was carried out with ethanol for five minutes, followed by a post-cure time of ten minutes in post-cure station (EDG Soluções, São Carlos, São Paulo, Brazil)

-The specimens were washed, dried, and after two minutes a thin layer of adhesive (Kulzer Mitsui Chemical Group, São Paulo, São Paulo, Brazil) was applied and polymerized for 1 minute and thirty sec, in a post-curing station (Kulzer Mitsui Chemical Group).

-Then, a layer of approximately 0.2 mm of R50 (Kulzer Mitsui chemical group)

-The same was done for the pink pigment (Kulzer Mitsui chemical group).

-To prevent the formation of the dispersion layer, 0.5 mm thick insulating gel (Kulzer Mitsui chemical group).

-After washing, a thin layer of glaze and light-curing sealant (Megadenta Dentalprodukte GmbH, Ribeirão Preto, São Paulo, Brazil) was applied and, after twenty sec, cured for five minutes, in a polymerizer (Kulzer Mitsui Chemical Group)

-Each specimen was identified on the side with a marking made by a drill (Labordental Ltda, São Paulo, São Paulo, Brazil).

-Mechanical brushing was performed on a Pepsodent machine (Acess. e Serv. Ltda. ME, Ribeirão Preto, São Paulo, Brazil)

-The specimens were brushed with a soft brush (Johnson & Johnson, São José dos Campos, São Paulo, Brazil)

-A portable colorimeter (BYK-Gardner, Geretsried, Germany) was used for color measurements with D65 standardized lighting within the visible spectrum (400 to 700 nm)

-The measurement of Knoop microhardness was performed with a Shimadzu Microhardness Tester (Shimadzu Corporation, Kyoto, Japan)

-For the measurement of surface roughness (ΔRa) a rugosimeter (Mitutoyo, Tokyo, Japan)

-The flexural strength assessment (EMIC, São Jose dos Pinhais, Paraná, Brazil) was verified at a crosshead speed of 5 mm per minute.

-For qualitative analysis of the surface, the specimens were positioned parallel to the table of the confocal microscope Olympus LEXT OLS4000® (Tokyo, Honsu, Japan)

-Samples were sterilized by hydrogen peroxide plasma (Advanced Sterilization Products, Irvina, CA, USA).

-The plates were incubated at 37º C for 1 hour and 30 minutes under agitation at 75rpm (Scientific Equipment, Campinas, São Paulo, Brazil)

-Brushing was performed manually, by the same operator, with new soft brushes (Johnson & Johnson), sterilized under UV light for 20 minutes.

-The test tube/specimen set was sonicated in ultrasound (Altsonic, Ribeirão Preto, São Paulo, Brazil) at 40KHz, 200W for 20 minutes, shaken individually in a test tube shaker (Phoenix, Araraquara, São Paulo, Brazil) and 0.025 mL of the suspension was seeded in dilutions ranging from 100 to 10-3 in Petri dishes containing specific culture medium.
